# The Senescent Heart—“Age Doth Wither Its Infinite Variety”

**DOI:** 10.3390/ijms25073581

**Published:** 2024-03-22

**Authors:** Anupama Vijayakumar, Mingyi Wang, Shivakumar Kailasam

**Affiliations:** 1Cardiovascular Genetics Laboratory, Department of Biotechnology, Bhupat and Jyothi Mehta School of Biosciences, Indian Institute of Technology Madras, Chennai 600036, India; anupamavijaykumar@gmail.com; 2Laboratory of Cardiovascular Science, National Institute on Aging/National Institutes of Health, Baltimore, MD 21224, USA; mingyiw@grc.nia.nih.gov; 3Department of Biotechnology, University of Kerala, Kariavattom, Trivandrum 695581, India

**Keywords:** aging, senescence, cardiac fibroblasts, cardiac myocytes, vascular cells, DDR2, collagen, cardiac fibrosis

## Abstract

Cardiovascular diseases are a leading cause of morbidity and mortality world-wide. While many factors like smoking, hypertension, diabetes, dyslipidaemia, a sedentary lifestyle, and genetic factors can predispose to cardiovascular diseases, the natural process of aging is by itself a major determinant of the risk. Cardiac aging is marked by a conglomerate of cellular and molecular changes, exacerbated by age-driven decline in cardiac regeneration capacity. Although the phenotypes of cardiac aging are well characterised, the underlying molecular mechanisms are far less explored. Recent advances unequivocally link cardiovascular aging to the dysregulation of critical signalling pathways in cardiac fibroblasts, which compromises the critical role of these cells in maintaining the structural and functional integrity of the myocardium. Clearly, the identification of cardiac fibroblast-specific factors and mechanisms that regulate cardiac fibroblast function in the senescent myocardium is of immense importance. In this regard, recent studies show that Discoidin domain receptor 2 (DDR2), a collagen-activated receptor tyrosine kinase predominantly located in cardiac fibroblasts, has an obligate role in cardiac fibroblast function and cardiovascular fibrosis. Incisive studies on the molecular basis of cardiovascular aging and dysregulated fibroblast function in the senescent heart would pave the way for effective strategies to mitigate cardiovascular diseases in a rapidly growing elderly population.

## 1. Introduction

The phenomenon of aging has long fascinated mankind. The ‘Seven Ages’ of Man, set out perhaps sardonically by Shakespeare in *As You Like It* four hundred years ago, captivatingly capture the inexorable process of aging and the deprivations that define it. The passage of time has only heightened our interest in the catalysts that transform “*the infant, mewling and puking in the nurse’s arms*” into the soldier “*seeking the bubble reputation even in the cannon’s mouth*” and the “*lean and slipper’d Pantaloon, with spectacles on nose*” before the “*strange eventful history*” of life ends in “*mere oblivion—sans teeth, sans eyes, sans taste, sans everything*”.

Time heralds the onset of the aging process, leading to decline in function, disease, and ultimately, death [1,2,3]. Time-related changes in tissues and organs of the body occur universally across all life forms. However, it is fascinating that various organs and systems age at different rates, probably because of multiple “clocks” that are systemic aging drivers or clocks overlaid with organ- or tissue-specific counterparts within the whole-body system [4]. Differences in aging rates could also be because of the presence of different cell types in different organs. The key drivers of the aging process have been aptly identified as genomic instability, telomere attrition, epigenetic alterations, loss of proteostasis, deregulated nutrient sensing, mitochondrial dysfunction, cellular senescence, stem cell exhaustion, and altered intercellular communication [5,6]. Further, these hallmarks of aging are found to have three common traits: (i) they appear during normal aging; (ii) their experimental intensification accelerates aging; and (iii) their experimental reduction delays aging [5,6]. Robust evidence indicates that these cellular and molecular characteristics of aging apply to the heart as well, enabling investigators to conceptualise the core of cardiac aging and understand the underlying mechanisms, which in turn may spur research to evolve specific countermeasures by targeting these triggers.

## 2. The Aging Heart—An Overview

The heart is a complex, myogenic muscular organ that consists of four morphologically and functionally distinct chambers and is responsible for pumping blood by repeated, rhythmic contractions. The myocardium (middle layer of the heart), which constitutes the bulk of the heart wall, comprises two inter-dependent compartments, the parenchyma and the stroma [7]. The parenchyma consists of terminally differentiated myocytes that are responsible for the contractile function of the heart. The stroma consists of different cell types such as fibroblasts, vascular cells, endothelial cells, macrophages, and mast cells. Further, the stroma includes the extracellular matrix (ECM), which is considered “the molecular net” whose laxity can determine the availability and composition of the acellular components of the heart, including interstitial collagens, proteoglycans, glycoproteins, matrix turnover-related enzymes, and cytokines and growth factors [8].

As any other organ in the body, the heart tends to age, and plays a “losing game of catch up”, presumably by priming the cells to work harder to maintain normal pump function. Cardiac aging is marked by aging of the contractile apparatus of the heart, the coronary vasculature and cardiac fibroblasts, and the deposition and aging of the ECM [9,10,11]. Major alterations in cardiac cell structure, biochemistry, and physiology include cardiac myocyte enlargement, increased myocyte senescence, fibroblast hyperplasia, alterations in the turnover and physical properties of ECM proteins like collagen, augmented cardiac fibrosis, increased left ventricular (LV) mass-to-volume ratio and left atrial size, diastolic dysfunction, valvular degeneration, epicardial adipose tissue deposition, increased prevalence of atrial fibrillation and reduced maximal exercise capacity [10,12,13,14,15]. Further, with advancing age, significant adverse vascular changes occur in the myocardium. Vascular aging is associated with vascular dysfunction, arterial stiffening, calcification of the aortic valve leaflets [12], macro- and microvascular endothelial dysfunction and myocardial ischaemia [16]. Moreover, to overcome the increased load imposed by the aging vasculature and preserve systolic function and diastolic filling, the myocardium undergoes remodelling [11,17]. Importantly, vascular changes promote myocardial ischaemia and consequent myocyte death, and in turn, the surviving myocytes undergo hypertrophy in response to mechanical/neurohormonal stimuli or haemodynamic stress [12,16]. Ultimately, these structural changes and compensatory responses in the aged heart lead to functional alterations such as adverse modification of the contractile machinery of the heart [12]. Another critical cellular response to aging-related decrease in myocyte number is the activation of cardiac fibroblasts and associated synthesis and secretion of fibrillar collagens and matricellular proteins [18,19]. Over a period of time, progressive interstitial fibrosis due to excessive matrix protein deposition causes stromal expansion, ventricular stiffening, and compromised cardiac function.

While our understanding of the role of external factors (like air pollution, exercise, diet, and psychological stress) and intrinsic factors (like Telomere shortening, genetic predisposition, and epigenetic modifications) [9] in influencing the function of the heart is largely established, what still stands out as a glaring lacuna is the lack of clarity on the alterations in cellular and molecular mechanisms underlying cardiac aging and their consequences. Admittedly, such insights can lead to new frontiers of investigation, and ultimately, the identification of potential therapeutic targets, which is particularly important in view of the increased transition of the Baby Boomer generation into the elderly demographic. Deeper insights into the physiology and biochemistry of cardiac aging would hinge around a sound understanding of the contributions of the cardiac stromal and parenchymal constituents to the aging process.

This review presents an overview of senescence-related changes in the principal cellular constituents of the myocardium that disturb normal cardiac physiology and function and provides a snapshot of the underlying mechanisms (Figure 1). Importantly, it discusses the paradoxical scenario of cardiomyocyte loss on the one hand and excessive collagen deposition due to reduced degradation on the other, which derails the quantitative relationship between the parenchyma and stroma and promotes adverse structural and functional alterations in the aged myocardium (Figure 2). Finally, this review focuses on Discoidin domain receptor 2 (DDR2), a collagen-activated receptor tyrosine kinase predominantly located in cardiac fibroblasts that can potentially be targeted to prevent or minimise myocardial fibrosis (Figure 3).

## 3. Myocytes in Cardiac Aging

### 3.1. Increased Cell Death and Loss of Replicative Capacity

Myocytes account for about 65 to 80% of the total volume of the adult mammalian heart but, during aging, myocytes are lost in large numbers due to apoptosis. In fact, myocytes in the aged heart are susceptible to apoptosis, and as many as 30% of them are lost due to aging under normal conditions [20]. The molecular mechanisms that promote cardiomyocyte apoptosis are varied and the exact causes of myocyte apoptosis in the healthy aging heart remain largely unclear. It may result from altered patterns of gene expression in response to cellular stress or due to the accumulation of DNA mutations and protein misfolding that elicit endoplasmic reticulum (ER) stress-mediated activation of the caspase cascade [20]. Further, as the myocardium ages, factors such as increased oxidative stress, hypoxia, and inflammatory responses compromise the survival of cardiac myocytes. Mitochondrial dysfunction due to mutations, alterations in mitochondrial gene expression, and a reduced threshold for mitochondrial permeability transition pore induction also promote myocyte apoptosis, contributing to the aging heart phenotype [20]. These functionally impacted myocytes, with altered metabolic, ionic, and electrical properties, affect myocardial performance negatively.

Importantly, myocyte loss is not compensated for by proliferation of surviving myocytes. Being terminally differentiated cells in a postmitotic organ, myocytes proliferate to a limited extent shortly after birth but almost completely lose their replicative capacity in the adult phase, possibly due to the downregulation of pro-proliferative growth factors, signalling pathways and innervation [21,22]. While an inconsequentially low turnover of <1% in human cardiomyocytes is supposed to be beneficial for the homeostatic maintenance of cardiac structure and function, such a low or insignificant rate of cardiomyogenesis cannot compensate for age-induced loss of myocytes, which impairs cardiac function [23].

Studies by Oh et al. [24] point out that telomere shortening may be responsible for replicative senescence via downregulation of telomerase reverse transcriptase (TERT) and that telomere dysfunction is associated with greater sensitivity to apoptosis. However, their investigations show that forced expression of TERT can only delay cell cycle exit in the cardiac muscle but can induce hypertrophy in postmitotic cells and promote cardiac myocyte survival. This suggests that the intrinsic absence of adult cardiomyogenesis remains an intractable problem in countering the structural and physiological changes in the myocardium during aging.

### 3.2. Myocyte Hypertrophy

During aging, when myocytes decrease as a percentage of the total cell number and cell regeneration does not compensate for cardiomyocyte loss, cardiac hypertrophy emerges as the key adaptive mechanism to meet these demands [20,25,26]. Myocyte hypertrophy is characterised by an increment in cell size, increase in protein synthesis, and changes in sarcomere organisation due, possibly, to age-related aberrant expression of certain isoforms of regulatory proteins like troponins and tropomyosin and cytoskeletal components like titin and myosin-binding protein C, leading to reduced contraction, reduced mechanical response, and disease [27,28,29]. Angiotensin II (Ang II), endothelin-1 (ET-1), and insulin-like growth factor-1 (IGF-1) are some of the known neurohumoral factors that can induce cardiac hypertrophy via receptor tyrosine kinases or G-protein-coupled receptors such as Gαq [20,30]. These receptor signals converge upon calcium signalling that may alter intracellular calcium levels, leading to the activation of calcium-dependent protein kinases, phosphatases, or mitogen-activated protein kinases (MAPKs) to mediate the hypertrophic responses. These in turn activate the ERK1/2 MAPK (extracellular signal-regulated kinase 1/2 MAPK), calcineurin/NFAT (nuclear factor of activated T-cells), CaMKII/HDAC (histone deacetylase), and Protein kinase B signalling pathways and their downstream regulators [20,30,31,32]. Similarly, the deregulation of nutrient and growth signalling pathways and the mammalian target of rapamycin (mTOR) pathway is also found to play an important role in cardiac hypertrophy and aging [33,34].

Compensatory cardiomyocyte hypertrophy response enables the myocardium to overcome myocyte loss by normalising biochemical stress and transiently optimising cardiac pump function. However, age-related loss and the initial compensatory or adaptive enlargement of myocytes to maintain the functional demands of the heart eventually burden the surviving myocytes with increased mechanical load, turning the otherwise adaptive hypertrophy pathological and detrimental. Ultimately, these alterations lead to left ventricular thickening, asymmetric growth of the interventricular septum, and change in heart shape from elliptical to spheroid [11]. These changes have significant implications for cardiac wall stress and overall contractile efficiency and result in reduced stress tolerance and decreased cardiac function [20].

Additionally, ECM accumulation around hypertrophied cardiomyocytes promotes cell death by reducing nutrient supply, resulting in a mismatch between supply and demand [19]. Aging-associated alterations in the ECM, such as collagen fibre deposition around myocytes, can also impair the propagation and transmission of electrical signals, promoting cardiac arrhythmias [12].

### 3.3. Ca^2+^ Handling

Cardiac aging compromises myocyte contractility by negatively impacting calcium influx and deregulating Ca^2+^ SR storage, culminating in impaired intracellular Ca^2+^ homeostasis [35]. Aging is marked, even at an early stage, by significant alterations in the expression of myocyte proteins that are involved in cardiac contraction and relaxation. Some of the major proteins involved in calcium handling, like sarcoplasmic/endoplasmic reticulum Ca^2+^ ATPase 2a (SERCA2a) and RyRs (ryanodine receptors), undergo age-associated changes in gene expression [35]. The decay of transient Ca^2+^ is significantly prolonged with age due to a decrease in the expression of SERCA2a and over-activation of RyRs [9,35,36]. This is particularly significant in the elderly population because it influences therapeutic effects and time of intervention. Interestingly, findings by Saeed et al. [37] report that, during aging, Ca^2+^ transient may be different in different parts of the heart because of compensatory contractile mechanisms.

In the myocardium of rodents and atria of humans, an isoform shift in myosin heavy chain transcript or protein levels is observed with aging [20,38,39,40]. Transcripts encoding sarcomeric actin isoforms are significantly altered with aging [20,41]. Functional changes in SERCA2 and expression changes in sodium–calcium exchanger-1 (NCX1) may also impact myocardial relaxation with age [20,42,43]. Further, levels of transcripts encoding factors like angiotensin II type 1 receptor (AT1R), nuclear factor kappa B (NF-κB), and the endothelin-A (ETA) receptor, which are associated with many critical signalling pathways that impact the contractile machinery, alter with aging [20,44,45].

### 3.4. Mitochondrial Changes and Cellular Energetics

Cardiomyocytes are highly vulnerable to age-related mitochondrial damage, characterised by mutilated mitochondrial structure and deficient ATP production [46]. The age-related changes in the mitochondria of cardiomyocytes can contribute to cardiac dysfunction, either by directly damaging the cellular macromolecules via oxidative damage or indirectly by interfering with normal cell signalling and cellular energetics, possibly through electron leakage [46,47,48]. Cardiac aging also adversely reprogrammes the metabolic activities of the cell, with immense consequences for cardiac function [49]. For example, under normal physiological conditions, mitochondrial ATP generation is mainly via fatty acid oxidation [49,50]. While lipoprotein- or adipose tissue-derived free fatty acids are the key source of cardiac ATP, glucose derived from circulation or from the hydrolysis of intracellular glycogen stores is used as the main fuel for energy production by cardiomyocytes in the foetal heart [49]. However, as the myocardium ages, the utilisation of fatty acids is reduced, and energy production happens at the expense of higher glucose oxidation, leading to the accumulation of toxic lipids like higher diacylglycerols and ceramides in the heart, which promotes cardiac lipotoxicity and associated complications [49,51,52,53,54]. Further, insulin resistance in the elderly population may also act as a checkpoint for glucose utilisation by modulating tissue uptake of glucose [55]. Although it may take longer for aged hearts to adapt or switch from glycolysis to ketone oxidation, ketone bodies might be a better alternative fuel in failing hearts since they can provide the collateral benefit of protection against oxidative stress in the aged heart [56,57]. A strategy that can enhance the oxidation of fatty acids and ketone bodies and limit carbohydrate metabolism in the aged myocardium would be of considerable therapeutic value.

### 3.5. Impairment of Cardiac Pacemaker Activity with Aging

Heart rate is regulated through neural control of the cardiac pacemaker. The sinoatrial node (SAN), a specialised tissue located at the right upper chamber (atrium), acts as the cardiac pacemaker and spontaneously generates electricity to initiate each heartbeat. There is a natural slowdown of the intrinsic pacemaker rate as age advances, which is the principal cause of pathological SAN dysfunction that necessitates artificial pacemaker implantation in humans [58]. As noted by Choi et al. [59], the slowdown of the pacemaker could be due to a combination of mechanisms, including the reduction in the activation of hyperpolarisation-activated cyclic nucleotide-gated (HCN) channels, loss of pacemaker cells, and an increase in tissue fibrosis. Noradrenaline, released from sympathetic terminals, activates β1- and β2-adrenergic receptors (ARs) on the plasma membrane of pacemaker cells. L-type calcium channels are a major downstream target of β-AR activation. Aging also leads to a reduction in the response of pacemaker cells to noradrenaline [60,61]. Notably, it is reported that aging does not alter adrenergic regulation mediated by β1-ARs but severely impairs that mediated by β2-ARs [59].

## 4. Role of Non-Myocyte Cells in Cardiac Aging

In addition to functional impairment resulting from the aging of myocytes, aging of non-myocytes can also interfere with cardiomyocyte aging and cardiac function, which necessitates a close examination of the non-myocyte population in the aging heart [9].

### 4.1. Vascular Changes

Non-myocytic cells in blood vessels—Vascular Smooth Muscle Cells (VSMCs), endothelial cells, and pericytes—proliferate to restore cardiac function after injury. However, as the vasculature ages, these cells acquire a senescence-associated phenotype or phenotypic switching towards proliferation, migration, and apoptosis [62]. The critical determinants of vascular aging that compromise vascular function include oxidative stress, DNA damage, chronic low-grade inflammation, epigenetic alterations, telomere attrition, mitochondrial dysfunction, deregulated nutrient sensing, defects in protein processing, reduced stem cell availability, and cellular senescence, all of which can result in vascular dysfunction associated with inflammatory response and intimal thickening, which is exacerbated by injury [63,64,65]. Further, mitochondrial dysfunction involves reduced ATP and increased reactive oxygen species (ROS) generation, which are detrimental to vascular function [64]. ROS generation causes the inactivation of nitric oxide (NO), which may adversely impact NO-dependent vascular relaxation, vascular cell proliferation and the anti-thrombogenic action of NO [64,66]. ROS also damages proteins, lipids, and DNA through oxidative post-translational modifications and triggers the NF-kB-mediated release of cytokines and other inflammatory mediators, leading to chronic low-grade inflammation [64,67]. Telomere attrition and uncapping (the breakdown of their loop structure) and epigenetic alterations are also among the factors that promote vascular dysfunction, vascular homeostatic imbalance, and pathological remodelling [68,69,70]. Mechanical stress, which promotes the destruction of elastic fibres, is another factor that exacerbates vascular aging, and age-related changes in the ECM render vascular cells more vulnerable to mechanical stress [64].

In the vascular media and adventitia, VSMCs and adventitial fibroblasts act as the principal source of collagen types I and III, maintaining vascular structure and function [71]. These cells facilitate adaptive structural modifications via ECM turnover to meet altered functional demands in response to haemodynamic modifications in the heart. However, prolonged exposure to pathological stimuli leads to excessive deposition of ECM proteins, particularly collagen [71,72,73]. Apart from cardiac fibrosis, vascular fibrosis and remodelling due to collagen accumulation aggravate with age [74], compromising vascular resilience and contributing to the onset and progression of atherosclerosis [75,76,77]. Age-related perturbations in immune functionality and the inflammatory response and age-associated build-up of atherosclerotic plaques, caused primarily by the accumulation of senescent cells and calcification within the coronary artery, may impair vascular structure and function, culminating in vascular diseases [78,79,80].

Changes in the intima and vascular endothelium also promote vascular dysfunction associated with aging. The thickened intima of aged rats shows augmented expression of transforming growth factor-β (TGF-β), Intercellular Adhesion Molecule 1 (ICAM-1) and matrix metalloproteinase-2 (MMP2). Accumulation of TGF-β in the intima of older rats can trigger the synthesis of ECM proteins such as fibronectin and collagen [81,82,83].

Additionally, dysregulated matrix metalloprotease activity and consequent degradation of elastin and increased collagen content bring about arterial stiffening and endothelial dysfunction [84,85,86]. Further, since fibronectin and TGF-β are under the regulatory control of Ang II, and angiotensin converting enzyme (ACE) and Ang II increase in the aged aorta, it is reasonable to postulate that the local renin–angiotensin system may be a key factor in age-related vascular remodelling [87,88,89,90]. Similarly, aging can disturb many metabolic and haemodynamic mechanisms in the vascular endothelium, making it senescent and dysfunctional [91,92]. With aging, endothelial cells lose their ability to proliferate. On the other hand, while the release of vasoactive molecules like NO and acetylcholine is reduced, the synthesis and release of ROS, endothelin, and Ang II are increased, resulting in a stiffened and hardened vessel wall that is susceptible to the development of many cardiovascular complications [91]. In animal models of aging, altered levels of extracellular enzymes like MMP-1 and tissue metallopeptidase inhibitor 1 (TIMP-1), matrix proteins like collagen type I, and growth factors like TGF-β and vascular endothelial growth factor (VEGF) profoundly affect endothelial cell proliferation and migration, negatively impacting the vital process of angiogenesis [93].

### 4.2. Pericytes in the Aging Heart

Pericytes, which are periendothelial cells that wrap around endothelial cells throughout the body, play an important role in the maintenance of stability and homeostasis of the vascular network. Although the effect of aging on pericytes remains obscure, there is increasing appreciation that delineation of pericyte responses to cardiac aging and disease might potentially pave the way for novel therapeutic strategies to reverse, or at least reduce, vascular structural remodelling. According to a recent report by Luxan et al. [94], gene ontology analysis of differentially expressed genes in pericytes of the aged heart showed that genes related to filopodia and actin cytoskeleton are upregulated while genes related to focal adhesion are downregulated. The study also found that Regulator of G-protein signalling 5 (RGS5), a repressor of G protein-coupled receptor (GPCR) signalling, is downregulated while RGS5 knockdown induced a contractile, pro-inflammatory, and pro-fibrotic gene expression profile, which diminished pericyte proliferation and migration. The study also reports that T-box transcription factor 20 (Tbx20), a cardiogenic transcription factor, is enriched in aged pericytes. Tbx20 is a repressor of platelet-derived growth factor receptor beta (PDGFRβ), the gene that encodes PDGFRβ and controls pericyte adhesion. The authors propose that RGS5 and T-box transcription factor 20 (Tbx20) may regulate pericyte function in the aged heart by controlling PDGFRβ signalling and cellular proliferation, adhesion, and migration.

## 5. Cardiac Fibroblasts and Aging

Cardiac fibroblasts are the principal stromal cells of the myocardium. They are undifferentiated cells accounting for more than 90% of the cardiac interstitial cells and about two-thirds of the myocardial cell population [95]. Apart from being the primary intracardiac source of the major fibrillar type I and type III collagens of the myocardium [8,95], cardiac fibroblasts are a “source and target” of several growth factors and cytokines that exert significant paracrine actions on co-resident myocardial cells [96,97,98]. Recent years have witnessed a surge in interest in the critical role of the cardiac fibroblast, which acts as a “friend” facilitating wound healing following myocardial injury but can also turn into a “foe” that goes into overdrive upon myocyte loss and causes excessive collagen deposition that can, alongside myocyte loss, upset the critical balance between the parenchyma and stroma, leading to cardiac dysfunction. Surprisingly, however, research over the years focused predominantly on the role of cardiac myocytes in aging, and cardiac fibroblasts were largely disregarded in terms of their role in cardiac aging and senescence.

### 5.1. Origin of Cardiac Fibroblasts

Cardiac fibroblasts are of heterogenous origin [99]. Fate-mapping studies, which uncover the developmental history of each cell in the adult body, suggest that cardiac fibroblasts are derived from different progenitor cells, depending on the cellular context and the stage of heart maturation [100,101]. The endocardium and epicardium are the two major distinct compartments that give rise to cardiac fibroblasts in the resting heart during embryonic development [102,103,104]. Furthermore, both of these compartments undergo epithelial-to-mesenchymal transition (EMT) to give rise to fibroblasts [102,103,104]. Fibroblasts isolated from homeostatic adult heart are mostly of epicardial origin. Majority of embryonic cardiac fibroblasts are derived from two principal sources: (1) the pro-epicardial organ and (2) EMT. Mesenchymal cells in the embryonic pro-epicardium give rise to a migratory cell population that migrates over the surface of the embryonic heart and forms the epicardium [101]. Under the influence of growth factors, some of these cells undergo EMT to become epicardial-derived cells (EPDCs) that invade the atrial and ventricular walls, differentiate into a fibroblast phenotype, and establish the compact myocardium [105]. Nonetheless, upon injury to the adult heart, resident fibroblasts and cells from spatiotemporal sources such as endothelial cells, bone marrow-derived progenitor cells and neural crest, circulating monocytes, pericytes. or circulating progenitor cells also differentiate into myofibroblasts that facilitate tissue repair [100]. Additionally, upon exposure to pathological stimuli, new fibroblasts are formed from pre-existing fibroblasts rather than through de novo EMT [103,104,106].

### 5.2. Mechanisms of Cardiac Fibroblast-Mediated Wound Healing in the Myocardium

In the healthy heart, fibroblasts remain quiescent under the shielding effect of matrix proteins that protect these cells from mechanical stress and prevent their activation [19]. However, any kind of injury to the heart—mechanical, inflammatory, or ischaemic—causes the activation or differentiation of cardiac fibroblasts [99]. Interestingly, injury to the young adult heart and loss of myocytes signal the normally quiescent cardiac fibroblasts to make up for the loss through an intricate wound healing mechanism that involves three stages—the adaptive phase, inflammatory phase, and granulomatous phase [107]. In the early adaptive phase of the response, the MMP/TIMP balance is altered, which triggers changes in net proteolytic activity [108]. This enables ECM degradation and the infiltration of inflammatory cells into the site of injury to clear the wound by phagocytising dead myocytes and remnant cellular debris. Significant neurohumoral changes and a dramatic surge in cytokine production in the myocardium, mainly of Ang II and TGF-β, mark the inflammatory phase. Furthermore, myocyte injury promotes a significant spike in cytokine production that causes the activation of cardiac fibroblasts, setting the stage for myocardial repair. Prior to transformation into the myofibroblast phenotype, cells go through an intermediate phenotype called the “proto-myofibroblast”, which is associated with impaired expression of alpha-smooth muscle actin (α-SMA) and increased formation of stress fibres, which facilitate migratory capacity [109,110,111,112]. Essentially, cardiac fibroblasts undergo phenotypic transformation into activated myofibroblasts that are hypersecretory and are characterised by contractile property due to the expression of α-SMA. They migrate chemotactically from the wound margin to the zone of injury and converge on the site of myocyte loss, proliferate, and produce matrix proteins to replace the damaged myocytes and facilitate healing [96,113]. This is the fibrogenic phase characterised by augmented production of ECM proteins such as collagen types I and III, fibronectin, and laminin, causing the granulomatous tissue to progress to a mature scar. This highly regulated process culminates in a condition called “reparative fibrosis”, which is mainly interstitial [114]. It should also be noted in this regard that conditions like inflammation or an increase in left ventricular pressure in the heart can initially lead to the expansion of cardiac interstitial space without any apparent loss of myocytes, which is a phenomenon called “reactive interstitial fibrosis” and is primarily perivascular [115]. Over time, this leads to myocyte loss and reparative or replacement fibrosis.

### 5.3. Impaired Wound Healing and Fibrosis in the Aged Myocardium

A striking pathophysiologic hallmark of the aging heart is cardiac fibrosis and associated fibrotic remodelling. Unlike conventional cardiac fibrosis, which is characterised by inappropriate proliferation and excessive ECM deposition, aging-related fibrosis is marked by degenerative changes with a rise in myocardial collagen content [116]. As age advances, even a healthy heart tends to progressively accumulate collagen, which leads to cardiac fibrosis and progressive increases in ventricular stiffness, culminating in impaired diastolic function [19,117]. However, very little is known about how transcriptional programs and fibroblast proliferation in the heart change with aging and impact these processes. Studies on animal models and human autopsy specimens showed that collagen content is increased during aging [118,119,120,121] while autopsy of 80-year-old people revealed an increase in collagen I and decrease in collagen III content in the heart compared to younger subjects [122]. However, earlier observations showed that transcript levels of collagen types I and III are either decreased or remain unchanged with cardiac aging [123,124]. Consistent with this finding is the report that mRNA levels of both procollagen type I and type III decrease, although the hydroxyproline content and histological quantification of LV collagen increase with age in rats [125]. In the pathogenesis of aging-associated fibrosis, reduced collagen degradation may be more paramount than enhanced de novo collagen synthesis [19]. It appears that elevated myocardial collagen in aging is most likely related to post-synthesis or degradative processes [124]. Enzyme activity, interactions among signalling pathways, protein synthesis and post-translational modification can also drive increased ECM deposition.

### 5.4. Dysregulated Collagen Turnover Promotes Excessive Collagen Deposition in the Aged Myocardium

A fine balance between the matrix-preserving and matrix-degrading signals determines the levels of myocardial collagen and is a *sine qua non* for the structural integrity of the heart [19]. Evidence suggests that diminished collagen synthesis and mRNA levels of collagen types I and III in the aged myocardium are associated with reduced levels and activity of MMP 1 and 2, suggesting that the attenuation of matrix-degrading pathways may be the dominant cause of excessive collagen deposition in the aging heart [19,126,127,128]. Moreover, the production of enzymes (prolyl hydroxylase, lysyl hydroxylase, and secreted protein acidic and rich in cysteine (SPARC), a matricellular protein) involved in the synthesis and processing of collagen fibrils by cardiac fibroblasts is also upregulated in the aging heart, leading to mature collagen deposition and ECM stiffness [19,124,126,127,128]. It is also to be noted that the level of non-enzymatic glycosylation or glycation of collagen increases gradually with aging, which can affect cardiac stiffness [129,130]. This is particularly important because collagen formation involves enzymatic processes that create crosslinks, promoted by glucose and other sugars [130]. Therefore, possible mechanisms underlying the increased collagen content in the aged heart are as follows: (1) reduced collagen degradation; (2) nascent collagen undergoes cross-linking with age such that collagen fibre stiffness increases, enhancing resistance to proteolysis by collagenases or MMPs; (3) as the matrix gets increasingly cross-linked, it may exert a negative feedback effect on genes regulating cross-linking; (4) collagen accumulation in the aged heart per se may exert a negative feedback effect on collagen gene expression in cardiac fibroblasts.

A study by Achkar et al. [131] demonstrated sex-dependent differential patterns of cardiac fibrosis and fibroblast phenotypes in aging mice. They showed that while reactive fibrosis in the myocardium and epicardium is seen in the aged hearts of female mice, perivascular and replacement fibrosis are observed in male hearts. Pappritz et al. [132] report that the collagen content of female cardiac fibroblasts is higher in 12-month-old animals. These observations point to the need for gender-dependent anti-fibrotic management in the elderly population. Moreover, cardiac fibroblasts from male animals produce higher levels of chemokines like CC motif chemokine ligands (CCL) 2 and 7, which shows their higher inflammatory capacity compared to their female counterparts.

### 5.5. Dysregulation of Cardiac Fibroblast Response to Injury in the Aging Heart

Apart from significant alterations in processes related to matrix turnover in the normally aging heart, dysregulation of the response of the aging heart to acute and chronic injury from a plethora of diverse insults contributes significantly to adverse structural and functional myocardial remodelling. As indicated earlier, the efficiency of wound healing after myocardial injury is determined by phenotypic changes in the fibroblast and its ability to mature into a functional myofibroblast. However, this efficiency gets compromised as the heart ages, which promotes adverse remodelling and jeopardises the outcome [133]. Apart from the effects of changes in cardiac fibroblasts on ECM, ECM tension can directly impact cardiac fibroblast phenotype and activation status via mechanosensing. It has been suggested that events occurring downstream of the matrix, such as activation or changes in expression levels of various proteins participating in mechanotransduction, can negatively alter the ability of the aging fibroblast to become a myofibroblast [134]. In their review, Angelini et al. [134] discuss the changes that occur with aging in ECM receptors (integrin or non-integrin), focal adhesions, cytoskeleton, and transcription factors involved in mechanosensing. In old male cardiac fibroblasts, the main effectors of the Kindlin/ERK/actin/α-SMA mechanosensing axis were recently reported to be defective [135].

Further, following injury, the myocardial repair machinery augments the number of mesenchymal stem cells (MSCs) or fibroblast precursors that can eventually differentiate into fibroblasts and mature into myofibroblasts [136]. However, MSCs derived from the aged heart give rise to fibroblasts with impaired function and differentiation. Not only is the transition of these cells from the fibroblast to the myofibroblast phenotype defective and they differentiate poorly into contractile myofibroblasts, reduced expression of TGF-β receptor I by the cells results in a blunted response to TGF-β and attenuation of the canonical TGF-β/SMAD pathway. This affects the migration of myofibroblasts to the area where replacement fibrosis is needed. The expression of α-SMA, a TGF-β-sensitive gene, is also reduced and myofibroblast differentiation is inhibited in the healing infarct [137]. All these lead to the formation of an insufficient scar after myocardial infarction. Trial and Cieslik [133] note that there is a paradoxical situation wherein reparative fibrosis is impaired but interstitial, adverse fibrosis is augmented. In an uninjured heart, the activated fibroblasts acquire a profibrotic phenotype causing interstitial fibrosis, ventricular stiffness and diastolic dysfunction, ultimately leading to heart failure [133].

Notably, JoAnn Trial et al. [138] have reported that pathologic fibrosis in the aging mouse heart is associated with dysregulated resident MSCs arising from reduced stemness and aberrant differentiation into dysfunctional inflammatory fibroblasts. Fibroblasts derived from aging MSCs secrete higher levels of collagen type 1 that directly contributes to fibrosis, MCP-1 that attracts leukocytes from the blood and IL-6 that facilitates transition of monocytes into myeloid fibroblasts.

It has also been demonstrated that the stimulatory effect of Ang II on collagen synthesis is less marked in rat fibroblasts isolated from senescent hearts, compared to fibroblasts harvested from young hearts [139]. It seems paradoxical that although the aged myocardium is exposed to increased oxidative stress [140], and oxidative stress-induced Ang II production would be expected to enhance collagen expression [141], there is actually a decrease in collagen synthesis in the aged heart due, at least in part, to defective response to Ang II. An ECM protein called periostin, an integrin-binding protein that is involved in cellular adhesion [142], alters the phenotype of fibroblasts through attachment-dependent signalling, which impacts myofibroblast differentiation. Periostin secretion in the heart can also affect the movement or adherence of fibroblasts to the site of myocardial injury [142,143]. Thus, periostin expression can be considered a marker of fibroblast activation and is involved in several aspects of fibroblast-mediated wound healing, including migration, differentiation, and collagen deposition. However, fibroblasts respond to periostin via α_v_ integrin, whose expression is found to be lesser in the infarct of an aging heart [143]. This suggests that myofibroblast differentiation may be compromised in the aged myocardium owing to reduced expression or availability of periostin.

### 5.6. Aging and a Poorly Organised Scar in the Injured Heart

It is pertinent to point out that a major feature of cardiac repair during aging is the production of poorly organised scar tissue, made primarily of collagen. Changes in collagen and ECM gene expression in the heart and in cardiac fibroblasts are maximal in the new-born and juvenile animal and decrease progressively with organismal aging [144]. Although excessive collagen accumulation marks the senescent heart, the efficiency of collagen deposition that normally facilitates efficient healing after MI is impaired with age, resulting in decreased tensile strength, increased dilation, and eventual dysfunction. Furthermore, the decrease in the deposition of collagen in the scar in the aged heart following injury can also be due to increased expression of MMP-9 without any apparent change in TIMP-3 expression [133]. Since a decrease in TIMP-3 activity is observed in failing hearts [145], and reduction in TIMP-3 expression is inversely correlated with MMP-2 and MMP-9 expression, the failure of TIMP-3 to increase and make up for elevated MMP-9 promotes collagen loss. Further, the altered responsiveness of fibroblasts from senescent infarcted hearts to factors like TGF-β exacerbates the reduction in the deposition of collagen from dysfunctional fibroblasts, resulting in significant alterations in ECM concentration and the formation of a scar with loose connective tissue. Apart from these, there is reduction in the expression of periostin and osteopontin and collagen deposition in the scar formed in the aged heart [137,146,147].

Finally, a variety of drugs, like anti-inflammatory agents, angiotensin-converting enzyme inhibitors, AT1R blockers, aldosterone and endothelin antagonists, and statins, may also affect healing, impair collagen deposition, and exert adverse effects on myocardial remodelling post injury in the aged population [148,149,150]. Thus, a plethora of senescence-related defects make the response of cardiac fibroblasts to myocardial injury less effective. In short, dysregulated and dissimilar collagen turnover mechanisms during the normal aging process and in response to injury may contribute to adverse structural alterations in the senescent heart.

Any discussion on aging-related changes in cardiac fibroblasts would be incomplete without a reference to fibroblast hyperplasia, which, alongside collagen production, is a major determinant of myocardial fibrosis post injury. In this regard, an age-dependent increase in cardiac fibroblast proliferation does not occur because the rate of cardiac fibroblast proliferation decreases during adulthood and plateaus through the rest of the organism’s life, indicating that phenotypic changes in the aging heart are not directly attributable to changes in the proliferative rate of cardiac fibroblasts [144]. On the contrary, changes in collagen turnover seem to be a more important cause of age-related changes in the cardiac interstitium.

### 5.7. Myocardial Aging and Impaired Immune Response to Injury

The immune system has an obligate role in myocardial repair following MI but its aging has significant implications for cardiac function in the elderly, in whom a compromised immune response to cardiac injury is observed [151]. Studies on old zebrafish have revealed that the immune system is activated in the ventricle, which is associated with impaired muscle organisation and regenerative response to myocardial injury [152]. The study showed that immune cells, mostly macrophages, accumulate in the old ventricle and show morphological and behavioural changes, impacting response to injury. It has recently been shown that age- and sex-dependent variations in inflammatory properties in cardiac fibroblasts lead to age- and sex-dependent differences in cardiac fibrosis and inflammation [132].

A striking feature of aging is systemic chronic inflammation, which is termed “inflammaging” [153,154]. It is different from the acute inflammatory response seen after MI, which is a low-grade immune reaction with no pathogen or injured tissue. It is suggested that the non-infarcted aged heart develops chronic inflammation partly because hematopoietic stem cells committed to the myeloid lineage preferentially expand over those committed to the lymphoid lineage within the bone marrow [151,155]. This in turn reduces overall lymphoid cell populations and amplifies myeloid cell populations that have become dysfunctional and promote cardiac inflammation [151,156,157,158]. Furthermore, the lymphatic system, which has a significant role post MI, declines with age [159]. Many immunological disorders, such as clonal haematopoiesis, myeloid skewing, and trained immunity, are age-related phenomena that can exacerbate adverse cardiovascular events. Finally, it is pertinent to note that, even as the link between cardiovascular health and immune cell activity is being re-examined, the underlying age-associated cellular changes remain largely unclear [151].

### 5.8. Molecular Changes in Fibroblasts with Aging

Significant alterations in the cell and molecular biology of cardiac fibroblasts with age have been reported sporadically. In fact, data from gene expression analyses suggest that differences in cardiac inflammation and immune response seen across developmental ages could be due to age-specific alterations in cardiac fibroblasts [160]. Transcriptomic analysis of the aged heart by single-nucleus RNA sequencing indicates that, among all cardiac cells, fibroblasts express most significant differential gene expression, increased RNA dynamics, and work entropy [161]. Apart from this, significant changes in the expression patterns of genes related to inflammation, ECM organisation, angiogenesis, and osteogenesis are observed in fibroblasts derived from aged animals.

An exquisite network of cellular crosstalk allows the healthy heart to function as an integrated unit with specific tasks assigned to sub-specialised cells. However, during aging, cardiac cells acquire an overt disordered state that in turn contributes to an altered cellular crosstalk, with significant implications for myocardial structure and function [162]. Functional analysis indicates impairment of paracrine interactions between fibroblasts and endothelial cells in older hearts. Increased secretion of Serpine 1 and 2 from older fibroblasts can exert antiangiogenic effects on endothelial cells [161]. Studies in vitro reveal that the anti-proliferative actions of C-type natriuretic peptide (CNP) on adult human cardiac fibroblasts is via the non- cyclic guanosine monophosphate (cGMP) pathway and the non-cGMP natriuretic peptide clearance receptor (NPR-C) antagonist attenuates the effects of CNP, suggesting that relative non-availability of CNP could be responsible for aging-related left ventricular fibrosis [163]. During aging, granulocytic myeloid-derived suppressor cells promote cardiac fibrosis by activating myofibroblasts and preventing senescence [164]. Further, given the role of Scleraxis (Scx) in regulating the cardiac fibroblast phenotype [165], it is possible that Scx may also play an important role in regulating mesenchymal character, conversion of fibroblasts to myofibroblasts, ECM synthesis, and cell contraction during aging.

## 6. Signalling Pathways in the Aging Myocardium

As noted in the preceding sections, age-associated aberrant signalling pathways and deficits in signalling in different myocardial cell types contribute to multiple aspects of myocardial pathophysiology, significantly impacting cardiac structure and function during aging. This review examines age-related changes in key signalling pathways in cardiac fibroblasts and their impact on altered matrix production and deposition in the aged heart.

### 6.1. The TGF-β Signalling Pathway in the Aging Heart

The TGF-β pathway is one of the major signalling cascades which is derailed during aging, leading to significant alterations in cardiac structure and function. TGF-β drives the activation of myofibroblasts and is induced under disease conditions and in response to certain stimuli [166]. Essentially a cytokine that binds to the TGF-β receptor to activate fibroblasts, TGF-β signalling orchestrates the synthesis and processing of ECM proteins, preserves the matrix by suppressing MMP activity and augmenting the synthesis of protease inhibitors such as PAI-1 and TIMPs, and increases α-SMA expression to promote fibroblast differentiation [19,167,168,169,170]. Interestingly, the decreased expression of TβRI in fibroblasts in the aged mouse heart, despite the increased availability of TGF-β in the heart, could be explained by the finding that prolonged TGF-β stimulation can downregulate the expression of both TGFβRs, namely activin receptor-like kinase (Alk) 5 and Alk 1 [133,136,171]. However, the increase in TGF-β levels in the aged heart, despite comparable levels of secretion of TGF-β by fibroblasts in young and aged mouse hearts, points to the probable secretion of TGF-β by infiltrating macrophages and T-cells [133,172]. Studies suggest that the latent form of TGF-β that is inaccessible to the receptors is released from ECM by the action of MMPs or integrin. Augmented MMP-9 expression, with a concomitant reduction in TIMP-3 expression, in the left ventricle of an uninjured aged heart may enhance MMP-mediated release of TGF-β from the matrix [133,173,174]. Similarly, the α_v_β_3_ and α_v_β_5_ forms of integrin bind to the latency-activated TGF-β peptide and liberate TGF-β from the complex via integrin-cell mediated force [175]. It is pertinent to point out that the existence of alternate signalling pathways that can facilitate ECM production, in the context of an apparent downregulation of TGF-β signalling in cardiac fibroblasts, cannot be discounted. In this regard, a crosstalk between fibroblasts from two different origins, myeloid and mesenchymal, that contributes to the development of fibrosis in the uninjured aging heart has been reported [172,176]. Due to defect in TGF-β receptor I (TGFβRI) expression, mesenchymal stem cells become unresponsive to TGF-β, leading to insufficient suppression of chemokine synthesis. This in turn causes an increase in mesenchymal stem cell differentiation and subsequent elevation in the number of collagen-secreting mesenchymal fibroblasts [172]. Furthermore, the lack of suppression of monocyte chemoattractant protein-1 (MCP-1) activates the influx of monocytes and their subsequent transition into macrophages and their differentiation into myeloid fibroblasts [176,177].

### 6.2. MAPK Signalling in the Aging Heart

MAPK signalling is another important modulator that links various adverse influences during aging, with detrimental effects on cardiac function. ERK1/2 MAPK signalling in cardiac fibroblasts is activated by TGF-β, PDGF, and EGF and has been found to be responsible for fibrosis [133]. In fibroblasts from the aged heart, activation of Ras-ERK-dependent signalling causes the upregulation of collagen [176]. Similarly, augmented levels of insulin cause further activation of ERK-dependent collagen type I expression in aged mouse cardiac fibroblasts. Another likely reason for enhanced ERK activation during aging is the down regulation of its negative regulator, namely, phosphatases [133]. A potent inducer of ERK1/2 MAPK activity is oxidative stress, which is a characteristic feature of aging as demonstrated by high levels of ROS in cardiac fibroblasts derived from the aged heart. Anupama et al. [141] have shown that, apart from triggering local Ang II production in cardiac fibroblasts, oxidative stress significantly increases collagen expression in cardiac fibroblasts via the ERK1/2 MAPK pathway. Similarly, in aged fibroblasts, increased ROS signalling may activate ERK1/2 MAPK and the expression of various cytokines and collagen [178,179].

p38 MAPK, another member of the MAPK family, has been found to enhance cardiac stress and promote fibrosis in aged mice following myocyte apoptosis in response to endoplasmic reticulum stress [180]. Further, blocking p38 MAPK signalling may alleviate oxidative stress and age-related fibrosis and other degenerative changes by inhibiting a ROS-sensitive kinase called “apoptosis signal-related kinase” [181]. Reduced expression of activin receptor-like kinase (Alk 5), a form of TβRI, causes impairment of the TGF-β-TAK1 (TGF-β-activated kinase 1)-p38 MAPK pathway in aging, thereby altering myofibroblast maturation [133,136]. The downregulation of dual-specificity phosphatases (DUSP1) during aging is also responsible for alterations in p38 MAPK activation and consequence [182,183]. The beneficial effects of inhibited MAPK signalling in mediating the actions of scutellarin, rosmarinic acid, and phosphocreatine are in line with earlier observations on the role of MAPKs in fibrosis and aging [184,185,186].

In this regard, various studies show that ROS activates the redox-sensitive transcription factor, NF-κB, via the MAPK pathway. Interestingly, NF-κB is considered to be the transcription factor most importantly associated with aging [187]. Furthermore, studies on animal models reveal that loss of Nfkb1 leads to premature animal aging, associated with reduced apoptosis and increased cellular senescence. Furthermore, loss of p50 DNA binding is a prominent feature of aged mice relative to young [188]. Similarly, nuclear factor erythroid-2-related factor-2 (Nrf2) is yet another redox-sensitive transcription factor that triggers the upregulation of anti-oxidant genes [189]. Exacerbated production of ROS in aging vessels tampers with Nrf2 activation, thereby increasing the sensitivity of blood vessels to the deleterious effects of ROS [190,191].

### 6.3. The Renin–Angiotensin–Aldosterone Pathway

Another pathway that is involved in fibroblast activation and is affected by aging is the renin–angiotensin–aldosterone (RAAS) pathway [192]. Ang II, as the effector peptide of the renin–angiotensin system (RAS), activates a number of intracellular signal transduction pathways via the AT1 receptor [193]. Apart from the circulating/systemic RAS cascade, the identification of a localised RAS in the heart with all components required for Ang II synthesis has unravelled the crucial role of this system in cardiac growth and development [194,195]. Interestingly, fibroblasts are the major RAS-positive cells in the heart, with very high AT1 receptor density. Increases in Ang II levels and AT1 expression resulting in enhanced RAS activity have been demonstrated in aged hearts and can lead to the activation and proliferation of fibroblasts, increased collagen deposition, and fibrosis [196,197]. It has been reported that senescent rat cardiac fibroblasts express the renin, angiotensinogen, and AT1 receptor transcripts, which impacts their response to Ang II [139]. Ang II can also act as an amplifier of other cytokines and growth factors, like TGF-β, whose expression is elevated in the aged heart. Moreover, Ang II crosstalks with the ERK1/2 MAPK, p38 MAPK, and TGF-β pathways. Aldosterone, another RAAS effector, is reported to be involved in cardiac fibrosis by directly affecting cardiac cells, especially fibroblasts. Treatment of isolated cardiac fibroblasts with aldosterone leads to a marked increase in cell proliferation via MAPK 1/2 activation [198,199,200]. Aldosterone-induced cardiac fibroblast migration is under the regulatory control of tumour necrosis factor receptor-associated factor 3 Interacting Protein 2 (TRAF3IP2), an upstream regulator of IκB kinase (IKK) and Jun N-terminal kinase (JNK) [201]. The addition of aldosterone to adult rat cardiac fibroblast cultures promotes a matrix-synthetic phenotype and stimulates collagen synthesis via type I corticoid receptors [200,202]. Clearly, more incisive investigations would shed light on the role of aldosterone in the aged myocardium.

### 6.4. MicroRNAs in Aging

The past decade has witnessed a surge in research into the role of microRNAs (miRNAs) in aging, addressing the plausible roles of miRNAs in regulating cardiovascular aging and age-related cardiovascular diseases [203,204,205]. miRNAs are small non-coding double-stranded RNA molecules of approximately 18–25 nucleotides that regulate gene expression post-transcriptionally by suppressing target protein expression by promoting mRNA degradation or by translational repression in both physiological and pathological conditions. miRNAs are highly conserved during evolution and are thought to regulate up to 90% of human genes [206]. Their remarkable stability and ease of detection in body fluids facilitate their use as biomarkers for age-related cardiovascular diseases [207,208]. Several miRNAs are suggested to be differently expressed during cardiac aging, which can potentially regulate different cell types and pathways [207]. For example, miR-34a is reported to be induced in the aging heart and its in vivo silencing or genetic deletion reduce age-associated cardiomyocyte cell death [209]. In fibroblasts, miRNAs that are induced during aging seem to have an association with fibrosis [210]. In the aged mouse heart, a reduction in the expression of miR-18 and miR-19, which are members of the miR-17–92 cluster, increases the expression of genes like connective tissue growth factor (CTGF) and thrombospondin-1 (TSP-1) that are importantly involved in ECM remodelling. In turn, this leads to adverse fibrosis and reduction in heart function [211]. On the other hand, enhanced expression of miR-22 during aging promotes fibroblast migration and senescence, at least in part via proteoglycan mimecan/osteoglycin, contributing to fibrogenesis [212]. miR 21, which is elevated in the aging heart, exerts pro-fibrotic effects on cardiac fibroblasts via the ERK1/2 MAPK pathway upon injury [203,207,213]. Further, apart from being a central regulator of cardiac fibroblast function, miR 1468-3p, whose expression is augmented in the aging healthy heart, mediates pro-senescence and pro-fibrotic functions via the TGF-β_1_-p38 MAPK pathway [214].

## 7. Senescence-Associated Secretory Phenotype in Cardiac Aging

Despite their inability to proliferate and repair damaged tissues, senescent cells tend to accumulate in the heart as the organ ages and influence neighbouring cells and tissues by secreting soluble factors. The senescence-associated secretory phenotype (SASP) is a marker or hallmark of cellular senescence [215] that modulates the tissue microenvironment through various physiological and pathological effects [80]. SASP depends on the senescent cell type and consists of ECM components, vesicles, exosomes, miRNAs, ROS, pro-inflammatory cytokines, chemokines, matrix degrading enzymes, and proteases that can transmit or enhance aging signals and trigger inflammatory responses [80,216]. In the mammalian heart, the cardiac stem/progenitor cell compartment undergoes senescence with age and manifests a phenotype that can negatively impact the neighbouring cells, leading to loss of the proliferative capacity of otherwise healthy and cycling-competent cardiac stem cells that switch to a senescent phenotype. However, SASP can signal immune cells to remove the senescent cells, although the immune system, whose functionality is compromised with increasing age, fails to do so [217,218]. While senescent cardiomyocytes release SASP factors that cause cardiac remodelling and dysfunction, cardiac fibroblasts undergo apoptosis or develop a senescent-like state with SASP expression upon continuous stress. Furthermore, IL-33 released by senescent fibroblasts can reduce cardiomyocyte senescence after hypoxic injury [219,220]. These fibroblasts are present abundantly in fibrotic areas, where they play a role in fibrotic myocardial pathologies. The SASP factors can also trigger the onset of many cardiovascular diseases or aggravate them. Thus, removing these senescent cells either genetically or via senolytic methods seems a promising strategy to ameliorate geriatric syndromes [220].

## 8. Discoidin Domain Receptor 2—A Potential Therapeutic Target to Mitigate Cardiac Fibrosis

The aforementioned facts unequivocally identify cardiac fibroblast-mediated myocardial fibrosis as a major contributor to cardiac dysfunction, regardless of age. Clearly, identifying cardiac fibroblast-specific mechanisms or factors that can be therapeutically targeted to prevent or minimise myocardial fibrosis is of immense scientific interest and clinical relevance.

In this regard, DDR2 is involved in a plethora of cellular processes, including matrix production, cell proliferation and cell death [221,222,223,224]. Several investigators have demonstrated its role in tissue fibrosis and cancer [224,225,226]. Recently, it was found that augmented expression of DDR2 within adventitial fibroblasts and VSMCs correlates with increased deposition and remodelling of collagen within the medial and adventitial layers of the abdominal aorta in a rhesus monkey model of metabolic syndrome [227]. Further, diet-induced collagen type I deposition and remodelling were attenuated by resveratrol, with a concomitant reduction in DDR2. Further, hyperglycaemia was found to increase DDR2 and collagen type I expression in isolated rat vascular adventitial fibroblasts and VSMCs via TGF-β1/SMAD2/3 signalling, which was attenuated by resveratrol. Gene knockdown and overexpression approaches confirmed an obligate role for DDR2 in hyperglycaemia-induced increase in collagen type I expression in these cells. The observations suggest that DDR2 may act as a molecular link between metabolic syndrome and arterial fibrosis, which would make it a potential therapeutic target.

In cardiac fibroblasts, Ang II has been shown to stimulate DDR2 expression via the redox-sensitive transcription factor, NF-kB. DDR2 in turn is reported to have an obligate role in Ang II-stimulated α-SMA and collagen gene expression in these cells [228,229]. Further, in cardiac fibroblasts exposed to Ang II, DDR2 increases the expression of anti-apoptotic cIAP2 that protects these cells against apoptosis [230], which can facilitate not only their role in wound repair in the short-term but also their persistence in an active form in the long-term, leading to excessive collagen deposition post injury. DDR2 also has an indispensable role in cell cycle progression in cardiac fibroblasts through transcriptional and post-translational mechanisms [230], as shown in Figure 3. Interestingly, a DDR2-dependent paracrine effect of cardiac fibroblasts on integrin-β1 expression in H9c2 cells has also been reported [229]. With its obligate role in the phenotypic transformation of cardiac fibroblasts and in the regulation of cell cycle progression, collagen and fibronectin expression, and apoptosis resistance in cardiac fibroblasts, DDR2 emerges as a master switch that holds the key to adverse cardiac fibrosis and myocardial remodelling. The critical role of DDR2 in cardiac fibrosis as a downstream target of Ang II, the predominant localisation of DDR2 in fibroblasts in the heart and the widely recognised efficacy of Ang II-targeting drugs in limiting adverse myocardial remodelling provide a compelling rationale to explore the Ang II-DDR2 link in relation to fibrogenesis in the normal and injured aged heart. It is tempting to postulate that such studies may lead to novel strategies that target DDR2 to specifically control cardiovascular fibrosis in the senescent heart (Table 1).

## 9. Summary and Future Perspectives

Oscar Wilde famously remarked, “With age comes wisdom, but sometimes age comes alone”! Be that as it may, aging invariably comes with nearly irreversible and deleterious structural and functional changes in the cardiovascular system. Even when adjusted for age-associated cardiovascular risk factors, aging per se emerges as an important contributor to heart failure. The steadily expanding elderly segment of the global population, saddled with the burden of cardiovascular disease, underscores the need to advance our understanding of what triggers and sustains the transition of gene expression patterns to give rise to a functionally compromised phenotype as age advances. Indeed, the identification of appropriate drug targets would hinge around such knowledge.

Until the 1990s, cardiac dysfunction was viewed mainly in terms of changes in cardiomyocytes, and incisive research exploiting tools of modern molecular biology has yielded valuable insights into critical aspects of myocyte pathophysiology. In time, attempts to bridge the translational divide led to the use of stem cells and other tissue engineering strategies, including 3D bioprinting, to achieve myocardial regeneration when cardiac progenitor cells fail to differentiate into myocytes post injury. There has also been considerable focus on ways to promote angiogenesis in the heart to address myocardial ischaemia. However, currently available methods to promote cardiac regeneration and angiogenesis leave much to be desired.

Notably, the recognition that dysregulated cardiac fibroblast function can upset the quantitative relationship between the parenchyma and stroma in the adult heart and compromise the structural and functional integrity of the organ has put the spotlight on cardiac fibroblasts and their contribution to the onset and progression of heart failure. Despite heightened interest in the cell and molecular biology of cardiac fibroblasts, attempts to evolve clinically viable approaches to prevent or minimise cardiac fibrosis have not picked up momentum commensurate with the severity of the problem. Therapeutic options to mitigate cardiac fibrosis remain limited largely to the gold-standard of blocking Ang II action [231,232], the overriding reason being the glaring gaps in our knowledge of the mechanistic basis of adverse fibrotic remodelling of the heart in diverse pathophysiologic contexts, including aging. It has been suggested that targeting molecules like osteopontin may be beneficial in the context of myocardial aging since it can modulate cardiac structure and function via its profibrotic properties [233]. Senolytics, which are compounds that can selectively induce senescent cell apoptosis [234], have lately drawn attention as antisenescence agents that may delay aging and attenuate age-associated adverse changes like cardiac fibrosis, thereby improving cardiac function. These drugs target components of the anti-apoptotic pathway, like B-cell lymphoma 2 (Bcl2), whose expression may be altered in senescent cells [235,236,237]. The best-studied senolytic drugs related to cardiovascular diseases include Navitoclax (ABT-263) and a combination of dasatinib, a tyrosine kinase inhibitor, with quercetin [234,238,239]. Yet another senolytic drug that is of immense interest is digoxin, a cardiac glycoside [220]. Senotherapeutics, which have been developed to pharmacologically eliminate the effects of senescent cells by suppressing SASP and other markers of senescence or by attenuating their pathological pro-inflammatory secretory phenotype to cause senostasis, are termed senomorphics [240]. Studies on microtissues developed by heart-on-a-chip technology that incorporates human-induced pluripotent stem cell-derived cardiomyocytes and cardiac fibroblasts have shown that treatment of fibrotic tissues with the senolytic drugs dasatinib and quercetin improves contractile function, reduces passive tension, and downregulates senescence-related gene expression, an outcome that is superior to what could be achieved using standard-of-care drugs [241]. Notably, functional improvement is associated with a reduction in fibroblast density, but with no changes in absolute collagen deposition. This study points to the benefit of senolytic treatment for cardiac fibrosis and is expected to enable the further elucidation of cell-specific effects of senolytics and how they may potentially impact cardiac fibrosis and senescence.

In their recent review on cardiac fibrosis, Claridge et al. [242] highlight the challenges to be overcome as we endeavour to understand the extremely complex regulatory mechanisms involved in cardiac fibrogenesis and discuss the approaches that might potentially generate useful insights (Table 1). For example, fibroblast heterogeneity, leading to divergent responses to insult, and regional variations in fibrogenesis within cardiac sub-structures, as evidenced by increased vascular but decreased myocardial expression of ECM components following COVID-19 infection [243], make it extremely difficult to arrive at a unifying mechanism that would explain the pathobiology of cardiac fibrosis. It would be important and challenging to ascertain the impact of aging on fibroblast heterogeneity and regional variations in fibrogenesis.

On a positive note, there is hope on the horizon, and for good reason; the anticipation is that what is to come will be better than what has been. Single-cell sequencing as well as quantitative proteomic and transcriptomic data on critical processes such as ECM turnover, mitochondrial metabolism and energetics, ion channel homeostasis, beta-oxidation, and glycolysis are expected to generate useful insights into the intricate mechanisms that lead to global structural and functional changes in the heart and promote cardiac fibrogenesis, across age groups. The future may also witness greater focus on the role of post-translational and epigenetic modifications, RNA-binding proteins, and non-coding RNAs. With their unique signature in cardiovascular diseases, miRNAs hold promise as a potential drug target [244], at least in part, to alleviate the deleterious effects of aging on cardiac cells. However, more evidence-based studies are required to confirm the translational potential of miRNA-targeted therapies. Further studies on microenvironmental and inter- and intracellular signalling mechanisms and single-cell omics technologies-driven incisive investigations into fibroblast heterogeneity would, in times to come, uncover the intricate molecular basis of cardiac fibrosis in the senescent heart, hopefully paving the way for improved therapeutic strategies that target the complexity of fibrotic myocardial remodelling.

## Figures and Tables

**Figure 1 ijms-25-03581-f001:**
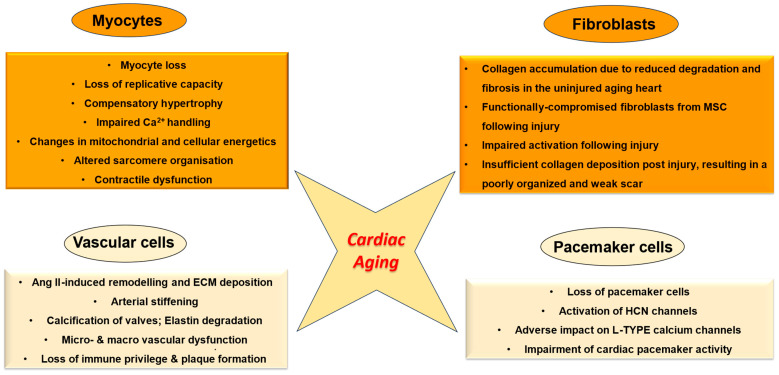
Schematic representation of senescence-related changes in the major cellular constituents of the myocardium.

**Figure 2 ijms-25-03581-f002:**
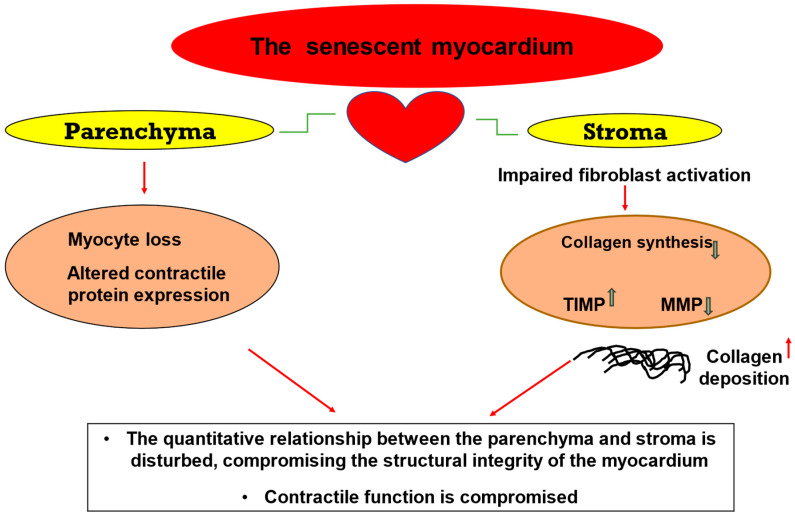
Disproportionate stromal expansion in the senescent myocardium. Myocyte loss and dysregulation of collagen turnover in cardiac fibroblasts promote adverse structural and functional alterations in the aged myocardium.

**Figure 3 ijms-25-03581-f003:**
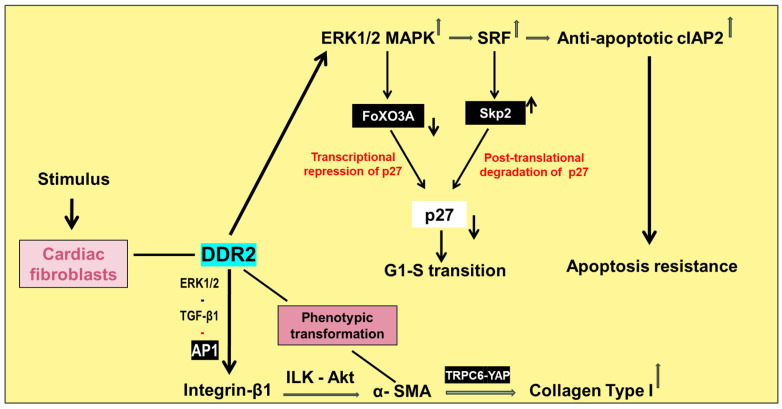
Is DDR2 the answer to cardiac fibrosis in the aging heart? DDR2 is a master switch in cardiac fibroblasts with an obligate regulatory role in phenotypic transformation, cell proliferation, collagen expression, and apoptosis resistance.

**Table 1 ijms-25-03581-t001:** A few challenges and potential solutions.

❖ Myocyte loss and failure of cardiac progenitor cells to differentiate into myocytes	➢ Improved tissue engineering approaches, including 3D bioprinting, for myocardial regeneration
❖ Adverse vascular changes promoting ischaemic conditions	➢ Promoting angiogenesis in the ischaemic myocardium
❖ Fibroblast heterogeneity that leads to divergent responses to insult and regional variations in fibrogenesis in cardiac sub-structures	➢ Probing microenvironmental and inter- and intracellular signalling mechanisms and single-cell omics technologies-driven incisive investigations into fibroblast heterogeneity to decipher the complexity of fibrotic myocardial remodelling
	➢ Targeting DDR2 to prevent or minimise cardiovascular fibrosis
❖ Mechanisms that trigger and sustain the transition of gene expression patterns, giving rise to a functionally compromised phenotype as age advances	➢ Generation of single-cell sequencing and quantitative proteomic and transcriptomic data on critical cellular processes across age groups
	➢ Focusing on post-translational and epigenetic modifications, RNA-binding proteins, and non-coding RNAs

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
