# Peer review of "The Senescent Heart—“Age Doth Wither Its Infinite Variety”"

_ijms, 2024, doi:10.3390/ijms25073581_

Round 1

Reviewer 1 Report

Comments and Suggestions for Authors

This review presents an overview of senescence-related changes and the underlying mechanisms during cardiac aging, a critical process which attracts enthusiastic attention recently. It discusses the changes of other principal cellular components of the myocardium besides cardiomyocytes, especially the cardiac fibroblasts. The review also focuses on a cardiac fibroblast-specific factor DDR2 which could be a potential target to prevent myocardial fibrosis. This review is well organized and well written, the structure is clear and easy to follow, the contents reviewed is thorough and up-to-date. There are a few minor points to improve this review.

1. The presentation of the figures needs to be improved, the impression is that they look like handouts for a class instead of figures in a paper. They are all very wordy, especially Figure 1 and Figure 4. The words in Figure 1 are so small and crowdy that are difficult to read, also not sure what differences between red and black fonds. It may be easier for the reader to comprehend the contents by reducing words in Figure 1 and convert Figure 4 into a table. For Figure 3, there is no symbol to indicate the organ or cell type this signaling cascade occurs.

2. I am confused about the saying DDR2 as a collagen-specific receptor tyrosine kinase, what does collagen-specific mean here? Collagen as the ligand for this RTK or the specific function of DDR2 is to regulate collagen expression? It needs to be clarified.

3. Some acronyms come without full words, for example VSMCs. Some appear multiple times, for example mesenchymal stem cells (MSC) appears twice. Nitric oxide and NO are both used, it needs to be consistent.

4. Line 147: Studies by [20]) suggest that, Line 307: According to a recent report by [84], Line 541: It is suggested by [139] that. These sentences seem missing the author names, as it appears in Line 426: A study by Achkar et al., 2020 demonstrated (but this one missing the citation).

Author Response

Response to comments

We thank the Reviewers for carefully evaluating the article and for their suggestions that have enhanced its overall quality. We have responded to all the comments and made the relevant changes that are highlighted in the revised manuscript for easy tracking.

We are happy to provide a point-by-point response to the comments.

Reviewer 1

Comments and Suggestions for Authors

This review presents an overview of senescence-related changes and the underlying mechanisms during cardiac aging, a critical process which attracts enthusiastic attention recently. It discusses the changes of other principal cellular components of the myocardium besides cardiomyocytes, especially the cardiac fibroblasts. The review also focuses on a cardiac fibroblast-specific factor DDR2 which could be a potential target to prevent myocardial fibrosis. This review is well organized and well written, the structure is clear and easy to follow, the contents reviewed is thorough and up-to-date. There are a few minor points to improve this review.

Response: We thank the Reviewer for the positive comments.

Comments

  1. The presentation of the figures needs to be improved; the impression is that they look like handouts for a class instead of figures in a paper. They are all very wordy, especially Figure 1 and Figure 4. The words in Figure 1 are so small and crowdy that are difficult to read, also not sure what differences between red and black fonds. It may be easier for the reader to comprehend the contents by reducing words in Figure 1 and convert Figure 4 into a table. For Figure 3, there is no symbol to indicate the organ or cell type this signaling cascade occurs.

Response: We are in total agreement with the Reviewer’s observations on the Figures. As suggested, Figure 1 has been modified substantially to summarize the principal alterations with age more clearly. The signaling pathways which were included in Figure 1 have been removed to eliminate clutter; these are anyway discussed in the text. As suggested, Figure 4 has been modified and presented as Table 1 to make it more comprehensible. Moreover, it has been organized in a way that it broadly presents the important questions to be addressed and the approaches that would help address them in future. In the modified Figure 3, the cell type where the signaling cascade occurs is indicated, in accordance with the observation of the Reviewer.

  1. I am confused about the saying DDR2 as a collagen-specific receptor tyrosine kinase, what does collagen-specific mean here? Collagen as the ligand for this RTK or the specific function of DDR2 is to regulate collagen expression? It needs to be clarified.

Response: DDR2 is described in the literature as a ‘collagen-specific receptor tyrosine kinase’ by many investigators. Nonetheless, in deference to the Reviewer’s comment, we have re-phrased it as “DDR2 is a collagen-activated receptor tyrosine kinase” in the revised Abstract and text.

  1. Some acronyms come without full words, for example VSMCs. Some appear multiple times, for example mesenchymal stem cells (MSC) appears twice. Nitric oxide and NO are both used, it needs to be consistent.

Response: We thank the Reviewer for drawing attention to this inconsistency. We have corrected these throughout the text. We have also ensured that all abbreviations are expanded at first mention.

  1. Line 147: Studies by [20]) suggest that, Line 307: According to a recent report by [84], Line 541: It is suggested by [139] that. These sentences seem missing the author names, as it appears in Line 426: A study by Achkar et al., 2020 demonstrated (but this one missing the citation).

Response: Again, we thank the Reviewer for pointing out these serious flaws. We have made the necessary corrections in these and other citations and ensured that all citations are in order.

Reviewer 2 Report

Comments and Suggestions for Authors

The review of Vijayakumar et al. focuses on the aging heart and especially on the role of fibroblasts. This overview is interesting, but I still have concerns regarding this review.

1) As the review focuses mainly on the role of cardiac fibroblasts during heart aging, this should be mentioned in the title.

2) There are some parts in the review which are not consistent. In line 105, the authors mention that proliferating cardiac fibroblasts deposit excessive collagen. In line 607-609, it is noted that ECM production leads to fibrosis in the aging heart. On the other hand, it is known that collagen/procollagen expression is reduced in the aging heart, as mentioned by the authors as well (lines 398-402). Therefore, the increase in collagen in the aging heart is mainly due to reduced collagen degradation.

Major reasons for this are accumulating non-enzymatic posttranslational modifications induced by oxidation and glycation, which are responsible for cross-linking. For example (10.1152/ajpheart.00168.2002) showed that glycation-mediated cross-linking plays a role in cardiac stiffness. Glycation can impair the proteolytic degradation of collagen by metalloproteinases (10.1096/fj.08-122648) and induce chronic inflammation (http://dx.doi.org/10.1016/j.redox.2013.12.016).

3) In the signaling part, the authors mention that during aging, ROS is increased and induces the MAPK pathway. If the authors believe that ROS is important, they should focus more on the NF-kB and the nrf2 pathways.

4) The sentence on SASP (lines 698-702) can be misleading as proinflammatory cytokines are part of the SASP and are not only a response to SASP.

5) Senolysis: there are some new papers published on the effect of senolysis on heart and vessel function, which should also be mentioned (lines 7156-716).

6) Figure 1 is complicated as it remains unclear why the authors mention specific changes in only one cell type. For example, reduction in telomere length is known for all models of replicative senescence (not only in endothelial cells), activation of cells is important for vascular cells as well, why only ECM deposition in vascular cells, and so on? Figure one has to be reconstructed, and the same structure (black/red parts) has to be used for all cells.

Author Response

We thank the Reviewers for carefully evaluating the article and for their suggestions that have enhanced its overall quality. We have responded to all the comments and made the relevant changes that are highlighted in the revised manuscript for easy tracking.

We are happy to provide a point-by-point response to the comments.

Reviewer 2

Comments and Suggestions for Authors

The review of Vijayakumar et al. focuses on the aging heart and especially on the role of fibroblasts. This overview is interesting, but I still have concerns regarding this review.

Comments

1) As the review focuses mainly on the role of cardiac fibroblasts during heart aging, this should be mentioned in the title. 

Response: Although the review lays relatively more stress on age-related changes in cardiac fibroblasts, changes in the other cell types are also covered, as noted in the last paragraph of the Introduction. Mentioning cardiac fibroblasts in the title without conveying the impression that the article is focused exclusively on cardiac fibroblasts is not easy and may even diminish the title. Moreover, the wider readership of the journal will be drawn more to a review on what appears from the title to be on the aging heart rather than to one that appears to be on the aging cardiac fibroblasts. So, with the reviewer’s kind concurrence, we prefer to retain the same title.

2) There are some parts in the review which are not consistent. In line 105, the authors mention that proliferating cardiac fibroblasts deposit excessive collagen. In line 607-609, it is noted that ECM production leads to fibrosis in the aging heart. On the other hand, it is known that collagen/procollagen expression is reduced in the aging heart, as mentioned by the authors as well (lines 398-402). Therefore, the increase in collagen in the aging heart is mainly due to reduced collagen degradation.

Major reasons for this are accumulating non-enzymatic posttranslational modifications induced by oxidation and glycation, which are responsible for cross-linking. For example (10.1152/ajpheart.00168.2002) showed that glycation-mediated cross-linking plays a role in cardiac stiffness. Glycation can impair the proteolytic degradation of collagen by metalloproteinases (10.1096/fj.08-122648) and induce chronic inflammation (http://dx.doi.org/10.1016/j.redox.2013.12.016).

Response:

We thank the Reviewer profusely for drawing our attention to inappropriate and inaccurate descriptions in a few places that point to jarring inconsistency. We have made the necessary corrections to remove those unintended contradictions. For example, in Line 105, we have removed “by proliferating cardiac fibroblasts” and changed the line to “Importantly, it discusses the paradoxical scenario of cardiomyocyte loss on the one hand and excessive collagen deposition due to reduced degradation on the other, which derails the quantitative relationship between the parenchyma and stroma and promotes adverse structural and functional alterations in the aged myocardium (Figure 2).” Notably, this modified line goes well with the Lines 398-402 pointed out by the Reviewer. Further, Lines 606-609 have been modified and now read, “It is pertinent to point out that the existence of alternate signaling pathways that can facilitate ECM production, in the context of an apparent downregulation of TGF-β signaling in cardiac fibroblasts, cannot be discounted.”

We have included the article (10.1152/ajpheart.00168.2002) in the manuscript. However, we feel that it is beyond the scope of this article to add (10.1096/fj.08-122648) since it is on how age-associated changes in extracellular matrix collagen can impair lung cancer cell migration. Further the other article, (http://dx.doi.org/10.1016/j.redox.2013.12.016) delves into the role of advanced glycation end products in cellular signaling.

3) In the signaling part, the authors mention that during aging, ROS is increased and induces the MAPK pathway. If the authors believe that ROS is important, they should focus more on the NF-kB and the nrf2 pathways.

Response: In deference to the Reviewer’s comment, we have included additional references on the NF-kB and the Nrf2 pathways, in the section on MAPK signaling in the aging heart. 

4) The sentence on SASP (lines 698-702) can be misleading as proinflammatory cytokines are part of the SASP and are not only a response to SASP.

Response: We thank the Reviewer for pointing it out. We have rephrased the sentence to dispel such an impression.

5) Senolysis: there are some new papers published on the effect of senolysis on heart and vessel function, which should also be mentioned (lines 7156-716).

Response: Again, we are grateful to the Reviewer for the suggestion as it enhances the value of the review. We have included additional recent references on this aspect.

6) Figure 1 is complicated as it remains unclear why the authors mention specific changes in only one cell type. For example, reduction in telomere length is known for all models of replicative senescence (not only in endothelial cells), activation of cells is important for vascular cells as well, why only ECM deposition in vascular cells, and so on? Figure 1 has to be reconstructed, and the same structure (black/red parts) has to be used for all cells.

Response: Both the Reviewers are unhappy with Figure 1 and we fully understand their objection. Accordingly, the Figure has been redone to highlight only the principal age-related alterations clearly, eliminating the issues noted by the Reviewer.